# Additional Treatments to the Local tumour for metastatic prostate cancer-Assessment of Novel Treatment Algorithms (IP2-ATLANTA): protocol for a multicentre, phase II randomised controlled trial

Martin John Connor [iD] ,[1,2] Taimur Tariq Shah,[1] Katarzyna Smigielska,[1,3] Emily Day,[3] Johanna Sukumar,[1,3] Francesca Fiorentino,[3] Naveed Sarwar,[4] Michael Gonzalez,[4] Alison Falconer,[4] Natalia Klimowska-Nassar,[1,3] Martin Evans,[1] Olivia Frances Naismith,[5] Kamalram Thippu Jayaprakash [iD] ,[6] Derek Price,[1] Shiva Gayadeen,[4] Dolan Basak,[4] Gail Horan,[6] John McGrath,[7] Denise Sheehan,[8] Manal Kumar,[9] Azman Ibrahim,[10] Cathryn Brock,[11] Rachel A. Pearson,[12] Nicola Anyamene,[13] Catherine Heath,[14] Iqbal Shergill,[15] Bhavan Rai,[16] Giles Hellawell,[17] Stuart McCracken,[18] Bijan Khoubehi,[19] Stephen Mangar,[4] Vincent Khoo,[20] Tim Dudderidge,[21] John Nicholas Staffurth [iD] ,[22,23] Mathias Winkler,[1,2] Hashim Uddin Ahmed [iD] [1,2]

For numbered affiliations see end of article.

**Correspondence to**
Mr Martin John Connor;
m.connor@imperial.ac.uk

## ABSTRACT

**Introduction** Survival in men diagnosed with *de novo* synchronous metastatic prostate cancer has increased following the use of upfront systemic treatment, using chemotherapy and other novel androgen receptor targeted agents, in addition to standard androgen deprivation therapy (ADT). Local cytoreductive and metastasis-directed interventions are hypothesised to confer additional survival benefit. In this setting, IP2-ATLANTA will explore progression-free survival (PFS) outcomes with the addition of sequential multimodal local and metastasis-directed treatments compared with standard care alone.

**Methods** A phase II, prospective, multicentre, three-arm randomised controlled trial incorporating an embedded feasibility pilot. All men with new histologically diagnosed, hormone-sensitive, metastatic prostate cancer, within 4 months of commencing ADT and of performance status 0 to 2 are eligible. Patients will be randomised to Control (standard of care (SOC)) OR Intervention 1 (minimally invasive ablative therapy to prostate±pelvic lymph node dissection (PLND)) OR Intervention 2 (cytoreductive radical prostatectomy±PLND OR prostate radiotherapy±pelvic lymph node radiotherapy (PLNRT)). Metastatic burden will be prespecified using the Chemohormonal Therapy Versus Androgen Ablation Randomized Trial for Extensive Disease (CHAARTED) definition. Men with low burden disease in intervention arms are eligible for metastasis-directed therapy, in the form of stereotactic ablative body radiotherapy (SABR) or surgery. Standard systemic therapy will be administered in all arms with ADT±upfront systemic chemotherapy or androgen receptor agents. Patients will be followed-up for a minimum of 2 years. Primary outcome: PFS. Secondary outcomes include predictive factors for PFS and overall survival; urinary, sexual and rectal side effects. Embedded feasibility sample size is 80, with

### Strengths and limitations of this study

- ► IP2-ATLANTA addresses an important research gap in the role of local and metastasis-directed therapy in men with newly diagnosed metastatic prostate cancer.
- ► This is the first phase II trial to include cytoreductive minimally invasive ablative therapy alongside cytoreductive radical prostatectomy and prostate radiotherapy.
- ► The IP2-ATLANTA study builds on the clinical benefits derived from metastasis-directed therapy (stereotactic ablative body radiotherapy and/or surgery) in a previously untreated cohort of men with advanced disease.
- ► Due to invasive interventions, blinding is not possible in the IP2-ATLANTA study.

918 patients required in the main phase II component. Study recruitment commenced in April 2019, with planned follow-up completed by April 2024.

**Ethics and dissemination** Approved by the Health Research Authority (HRA) Research Ethics Committee Wales-5 (19/WA0005). Study results will be submitted for publication in peer-reviewed journals.

**Trial registration number** NCT03763253; ISCRTN58401737

## INTRODUCTION

Overall, 47 000 men are diagnosed with prostate cancer each year in the UK.[1] Approximately, 4500 of these men this will be

diagnosed with *de novo* synchronous metastatic disease at presentation.[1] As with the USA, where just under 8% present with metastatic disease and where the annual burden is predicted to reach approximately 15 000 cases by 2025, so the prediction for the same magnitude is likely for the UK.[2]

Traditionally, such men were managed with androgen deprivation therapy (ADT) alone, via medical or surgical castration.[2] Unfortunately, the median time to the emergence of a castrate resistant state is in the order of 11–18 months, limiting overall survival (OS) to 3.5 years.[3 4] Promisingly, the reported OS in this group has now risen to a median of 4.8 years with the addition of upfront systemic agents, such as docetaxel, enzalutamide, abiraterone acetate or apalutamide.[5–9]

Moving beyond early systemic therapy escalation, there has been an increased focus on the role of local cytoreductive and metastasis-directed interventions (primarily stereotactic ablative body radiotherapy (SABR)) to gain additional oncological benefit.[2] This is in part based on the emergence of the 'oligo-metastatic state', which may exhibit different biological characteristics to poly-metastatic prostate cancer.[10]

Such men present with a clinically defined favourable metastatic burden and are hypothesised to occupy an intermediate state between 'locally advanced' and 'poly-metastatic' disease.[10 11] It is postulated, but not proven, that men exhibiting such disease patterns gain most benefit in progression-free survival (PFS) and OS resulting from localised cancer control achieved via cytoreductive interventions.[2 10]

### IP2-ATLANTA study hypothesis
We hypothesise that men with metastatic disease who undergo treatment of the local tumour in the form of either radical therapy (cytoreductive radical prostatectomy (CRP) or external beam radiotherapy (EBRT)), or minimally invasive ablative therapy (MIAT), combined with metastases-directed therapy (MDT), will improve PFS compared with patients who receive standard of treatment alone.

### Pathobiological basis for local cytoreductive and MDT
The pathobiological basis underpinning local cytoreductive and MDT in prostate cancer is not fully delineated.[2] Local prostate cytoreduction is thought to primarily impact on tumour-derived factors such as cytokines, chemokines and microRNAs.[12 13] In particular, prostate tumour cell shedding and dissemination has been shown to occur earlier, with the detection of circulating tumour cells (CTCs) in blood and disseminated tumour cells (DTCs) in bone marrow of patients staged as non-metastatic on conventional imaging (ie, bone scintigraphy).[14–18]

This has led to a comparison to the 'self-seeding' hypothesis, as described in other solid organ malignancies involving the breast and colon.[19 20] It posits that the return of CTCs or DTCs from distant secondary sites alters the primary tumour microenvironment, via release of matrix metalloproteinases (eg, matrix metalloproteinase-1) and cytokines (eg, CXC-motif chemokine 1).[19 20] Such circulation may lead not only to 'self-seeding' but also the remodelling of a 'pre-metastatic niche' at new distant sites.[19 21 22] Bone marrow–derived haematopoietic cells localise to support pre-metastatic niche's, promoting the local environment for colonisation.[14 16 22]

Furthermore, investigators using multifocal sequencing approaches have revealed the present of primary-tumour-to-metastasis, but also surprisingly, metastasis-to-metastasis transfer of clonal tumour cells.[23] This subsequently led to the exploration of metastasis-directed therapy.[24 25] Such interventions are hypothesised to have an effect on distant tumours via the release of tumour antigens, damage-associated molecular patterns and local activation of immune cells (including cytotoxic T-cells).[26–29] This has been coined the 'abscopal effect' and is associated with the generation of a systemic antitumour immune response. Evidence for such a response in prostate cancer remains sparse.[24 30]

Immune-mediate responses may not be limited to cytotoxic radiotherapy, with minimally invasive ablative therapy effects, such as the 'cryo-immunological response', also proposed.[31] Similar to the cytotoxic abscopal response, the clinical translation of such observed responses is unclear.[2] Early local prostate cryotherapy case series reported spontaneous distant regression of metastasis, although this has not been replicated in the contemporary literature.[31 32] Furthermore, clinical augmentation of prostate cryotherapy by immune-checkpoint inhibitors (eg, anti-programmed cell death-1 antibody, PD-1) also demonstrated preclinical promise, but proved disappointing when translated into early phase clinical studies.[33–35]

When taken collectively, removal of the primary tumour and possibly its metastatic sites may lead to a disruption in these immune-mediated pathological relationships and result in regression of metastases with a prolonged cancer-specific survival (CSS).

### Categorising metastatic burden
A key research barrier at present is that there is no universally accepted definition for oligometastatic disease, which varies depending on the anatomical site (nodal, burden, visceral), absolute number (1 to 7), spatial pattern (outside vertebral bodies or pelvis) and diagnostic imaging used (conventional or molecular).[11] Consequently, there is also no accepted definition for 'high' versus 'low' volume disease.[10 36 37] At present, oncology trials exploring systemic therapy have frequently adopted the use of the Chemohormonal Therapy Versus Androgen Ablation Randomized Trial for Extensive Disease (CHAARTED) definition of metastatic disease burden. High burden disease is defined as visceral metastasis and/or four or more bone metastases with at least one or more metastasis located outside the vertebral bodies or pelvis.[9 38]

## Cytoreductive prostate radiotherapy

Two randomised studies (STAMPEDE and HORRAD) have evaluated the role of cytoreductive local prostate radiotherapy in this cohort.[38 39]

The Systemic Therapy in Advancing Or Metastatic Prostate Cancer: Evaluation Of Drug Efficacy (STAMPEDE) collaborators explored the role of local prostate radiotherapy in 2061 men with newly diagnosed metastatic prostate cancer receiving ADT, with 18% also receiving docetaxel.[38] Although no OS advantage was demonstrated (HR 0.92 (95% CI 0.80 to 1.06); p=0.27) in all burden metastatic disease, radiotherapy did improve failure-free survival (HR 0.76 (95% CI 0.68 to 0.84); p<0.0001).[38] Nevertheless, in the prespecified subgroup of men with the CHAARTED definition of low burden disease, a significant OS was reported (3-year OS 81% vs 73%, HR 0.68 (95% CI 0.52 to 0.90); p=0.007). As with any subgroup analyses, these data need to be interpreted cautiously. Furthermore, radiotherapy treatment (weekly or daily) had acceptable side effects with only a 5% grade 3–4 adverse event rate.[38]

The HORRAD phase III trial randomised 432 men to ADT with or without local prostate radiotherapy.[39] In accordance with STAMPEDE, no significant difference in OS was observed between the two groups (median 45 months in experimental arm vs 43 months with ADT alone; HR 0.90 (95% CI 0.70 to 1.14); p=0.40), although there was a non-significant trend towards improved OS in the 160 patients with low volume metastatic disease treated with radiotherapy (HR 0.68, 95% CI 0.42 to 1.10). This study, however, was criticised for its lack of prespecified metastatic burden (including no knowledge of visceral disease) and potential underpowered sample size.[40] Both trials took place at a time when upfront systemic agents had not been fully introduced and thus the true 'additive' effect of local prostate radiotherapy in a contemporary cohort remains unclear.[2]

## CYTOREDUCTIVE RADICAL PROSTATECTOMY

Historical data from the Southwest Oncology Group (SWOG) 8894 trial randomising 1286 men with metastatic disease to bilateral orchidectomy with placebo or flutamide demonstrated that a subgroup of men who underwent previous radical prostatectomy had a significantly reduced risk of death (HR 0.77, 95% CI 0.53 to 0.80).[41] Building on this, numerous retrospective series and registry data (eg, Surveillance, Epidemiology, and End Results (SEER)) have reported improved OS and CSS in men who undergo cytoreductive radical prostatectomy with low-burden or predominantly osseous disease.[42–47]

At present, prospective evidence is limited to a 61-patient case–control study conducted by Heidenreich and colleagues.[48] Performed in men with <4 metastases and no visceral or extensive lymph node metastases, with a serum prostate-specific antigen (PSA) level <1.0 ng/mL after neoadjuvant ADT, CRP and extended pelvic lymph node dissection (ePLND) led to a 12.1-month improvement in PFS compared with the control arm ADT alone (38.6 months vs 26.5 months; p=0.0032).[48] There were no reported grade 4 or 5 Clavien-Dindo classification complications within this study, confirming the surgical feasibility reported in previous retrospective studies.[42 43]

Multiple confirmatory randomised (NCT01751438 (BST); NCT03655886 (LoMP II); ISRCTN15704862 (TRoMbone)) and single-arm studies (NCT02716974; NCT03298087) are ongoing with, or without, MDT in this cohort.[44 49–52]

## Cytoreductive MIAT

With regard to cytoreductive MIAT, a single retrospective study evaluating whole-gland cryotherapy in 23 men with a favourable response to 6-month ADT (PSA <1.0 ng/mL), </=cT3a disease, and limited bony metastasis, reported a 10-month survival advantage when compared with a matched cohort with ADT alone (35 months vs 25 months; HR 0.21 (95% CI 0.09 to 0.45); p=0.0027).[53]

Furthermore, the NCT02489357 pilot study interrogated the 'cryo-immunological response' with PD-1 blockade using the antibody (pembrolizumab) in addition to cytoreductive cryotherapy.[34] In total, 12 men with oligometastatic disease initiated 8 months of ADT and pembrolizumab, with subsequent whole-gland cryotherapy.[34] Primary endpoint was PSA <0.6 ng/mL at 1 year, and this was met in 42% (n=5). Median PFS was 14 months and median systemic therapy-free survival was 17.5 months.[34] There were no grade 3 adverse events, with grade 1 (non-pad, occasional) urinary incontinence in 16.7% (n=2).[34] This profile is in keeping with the favourable early functional outcomes from cryotherapy in patients with non-metastatic disease.[54] With regard to safety, there were no reported cases of rectal injury or fistulae in either study.[34 53] In both studies, men did not receive prior systemic therapy escalation and thus the 'additive' value of cytoreductive cryotherapy in such a cohort remains unclear.[34 53]

## Metastasis-directed therapy

In men with recurrent distant oligometastases, a number of early phase clinical trials (STOMP, ORIOLE, POPSTAR) have demonstrated promise with MDT (either SABR or metastasectomy), mainly with regard to improving ADT-free and early PFS.[24 55–57] The impact of such interventions on OS is unclear.[24 55 56]

In *de novo* oligometastatic disease, a single pilot study including 20 men underwent sequential systemic therapy (ADT), surgery (cytoreductive radical prostatectomy+PLND±RPLND) and consolidation SABR to visible bone metastasis.[58] A novel endpoint of 'undetectable PSA (</=0.05 ng/mL) following testosterone recovery' was used and achieved in 20% (n=4). However, 95% (n=19) achieved an undetectable PSA, irrespective of testosterone suppression, after all three treatments. The addition of SABR accounted for an undetectable PSA in 21% (4/19) of this subgroup of men, when treatments were analysed separately.[58]

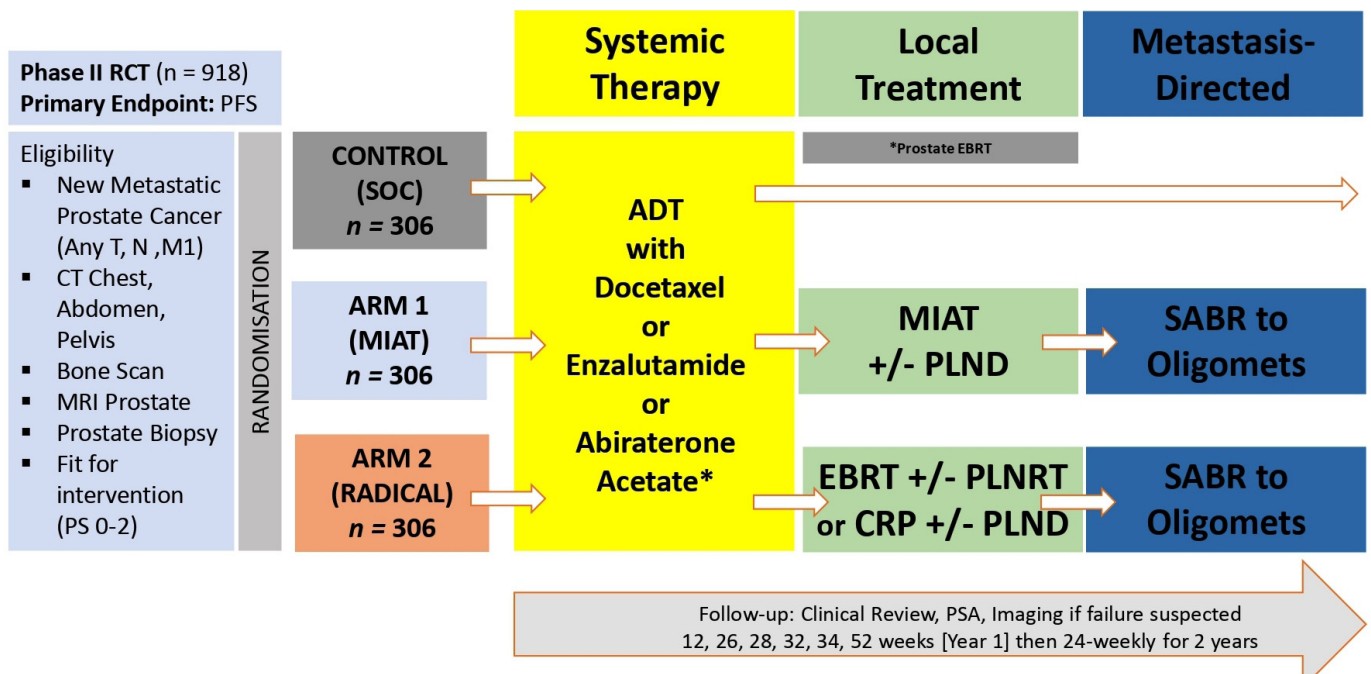

**Figure 1** Study flowchart. CRP, cytoreductive radical prostatectomy; EBRT, external beam radiotherapy; MIAT, minimally invasive ablative therapy; PLND, pelvic lymph node dissection; PLNRT, pelvic lymph node radiotherapy; RCT, randomised controlled trial; SOC, standard of care. *Systemic therapy is not limited to listed agents. *Prostate only EBRT may be performed in selected men with low-burden disease, if declared prior to randomisation and local SOC.

## METHODS AND ANALYSIS
### Study design and dates
IP2-ATLANTA is an unblinded, randomised, multicentre, interventional three-arm study with an active comparator arm incorporating standard of care (SOC) (figure 1). Study participants will be randomised to: Control arm (standard of care); Intervention arm 1 (MIAT±pelvic lymph node dissection (PLND)) or Intervention arm 2 (prostate external beam radiotherapy (EBRT)±PLNRT OR cytoreductive radical prostatectomy (CRP)±PLND). Systemic therapy in all arms includes ADT±docetaxel, abiraterone acetate, enzalutamide, apalutamide, as appropriate. Men with low-burden disease in intervention arms are eligible for MDT in the form of SABR or surgery (figure 1).

Study recruitment commenced in April 2019, with planned embedded feasibility recruitment completed by January 2021, due to severe COVID-19 recruitment impact. Planned main phase II component recruitment and follow-up is expected to be completed by April 2022 and April 2024, respectively.

### Patient and public involvement
A patient-involvement focus group was held with six patients who had advanced or metastatic prostate cancer to determine initial patient acceptability and gauge important opinions on the proposed amendment and study design. Four patients had previously received radiotherapy as either their primary or secondary treatment. Comments from the group discussion were recorded along with anonymous questionnaires, which the patients returned, by post after the meeting. Two patient and

public involvement representatives were present during the HRA REC assessment. They will continue to be involved throughout the duration of the trial with the Trial Management Group and other patients not involved in the direct management of the study will be on the independent Trials Steering Committee.

### Study population
Men who are willing to undergo local therapy to the prostate and selective MDT for newly diagnosed metastatic prostate cancer in addition to standard care systemic treatment upfront.

### ELIGIBILITY
#### Inclusion criteria
1. Diagnosed with prostate cancer within 6 months of screening visit.
2. Metastatic disease (any T, Any N, M1+) of any grade, stage or PSA level.
3. Fit to undergo SOC systemic treatment for metastatic disease and both minimally invasive therapy and prostate EBRT/cytoreductive radical prostatectomy.
4. Performance status 0–2.
5. Histologically proven local tumour.

#### Exclusion criteria
1. Patient did not undergo and/or is unable to undergo SOC baseline imaging tests for confirmation of metastatic status (CT abdomen/pelvis AND chest X-ray (or CT chest) AND radioisotope bone scan (or whole body

imaging such as MRI or Positron emission tomography (PET) imaging as alternative to all preceding scans mentioned here) AND prostate MRI.

2. Prior exposure to long-term ADT or hormonal therapy for the treatment of prostate cancer unless started within 4 months of screening visit.
3. Prior chemotherapy or local or systemic therapy for treatment of prostate cancer (apart from ADT or hormonal therapy as outlined above).

### Identification of patients

All men diagnosed with prostate cancer who go to a multidisciplinary team meeting or a tumour board, as well as any man meeting the eligibility criteria prior to tumour board discussion, will be identified for screening. Members of the tumour board will identify patients suitable for IP2-ATLANTA. The treating clinicians will mention the study and then the local research nurses/fellows, clinical trial coordinators, clinical trial practitioners or the treating clinicians will then approach the patients if they are interested. A Patient Information Sheet (PIS) will be given, or if agreed, emailed or posted out to the patient. Those patients already aware of the diagnosis can be approached by telephone to enquire as to their interest in the study so that a PIS can be then be sent out by email or post prior to a clinical visit. Patients will be given as much time as they need to read the PIS before consenting to participate (with a minimum of 24 hours).

### Randomisation

Stratified randomisation will take into account the following variables to create 16 strata in total:

▶ Intent to treat pelvic lymph nodes? Yes versus no.
▶ Metastatic burden (conventional imaging; CHAARTED definition)[9]: Low versus high.
▶ Intent to use systemic agent (ie, docetaxel, abiraterone acetate, enzalutamide, apalutamide)? Yes versus no.
▶ Intent to use metastasis-directed therapy? Yes versus no.

### Trial treatment
#### Control arm (SOC)

SOC systemic treatment regimen is determined by the treating clinician and will be declared upfront prior to randomisation. The decision as to which SOC systemic treatment is initiated should be made with reference to the current National Institute for Health and Care Excellence (NICE) and regional National Health Service (NHS) clinically commissioned guidelines.[59] At present, docetaxel is recommended for use in all men with newly diagnosed metastatic prostate cancer who do not have significant comorbidities.[59 60] Alternatively, new anti-androgen compounds, including but not limited to, abiraterone acetate or enzalutamide, are permitted if approved by regional NHS clinically commissioned guidelines.[59 61]

If radiotherapy is planned for local disease in men randomised to the SOC arm with low-volume metastases,

then this will be declared prior to randomisation by the treating clinician. For men with low burden disease, external beam prostate radiotherapy will be permitted and defined by the Local Radiotherapy Standard Operating Procedure (SOP) which reflects NHS clinically commissioned guidelines.[62] The use of PLNRT and/or MDT will not be permitted in the control arm. Palliative prostate radiotherapy for locoregional symptom control in men with high burden (>/=4) metastases will be permitted as per local clinical practice. Palliative bone radiotherapy for symptoms and prevention of fracture will be permitted in all men as per local clinical practice.

### Intervention arms

While discussing the intervention arms, we should also consider the impact that SOC systemic treatment may have on downstaging the local tumour. As the SOC systemic treatment would be administered prior to any local MIAT or CRP/EBRT in the intervention arms 1 and 2, an attempt at reclassifying the residual disease with a prostate MRI and biopsies would be pragmatic. This is to prevent patients from developing adverse events from unnecessary local treatment when they have no evidence of residual disease. Patients with positive post-SOC biopsies would then receive the local treatment as outlined below. Randomisation would occur at enrolment with planned intention-to-treat and per-protocol analyses.

Biopsies at 6–9 months from initiation of SOC systemic therapy are part of the protocol during the embedded feasibility pilot. As we currently do not know the significance of residual disease after SOC systemic therapy, even if determined to be low volume and low grade, all patients with positive biopsies will be offered local treatment as per randomisation. The pilot stage would obtain a point estimate of the magnitude of this response and also patient acceptability of a post-systemic therapy prostate biopsy.[63–66] Taken collectively, this will assist in informing the study investigators of their ongoing utility in the main phase II component, where they are not presently mandated.

### Intervention arm 1: MIAT

MIAT to the prostate with or without PLND in addition to SOC systemic treatment. The exact treatment protocol and modality used (cryotherapy or high-intensity focused ultrasound) will be set within the trial MIAT SOP. For those patients who are undergoing MIAT, no local prostate radiotherapy will be given as part of the intervention. Radiotherapy can be administered for palliative reasons. PLND will be performed based on the presence of resectable disease, patient fitness and consent/discussion with operating surgeon, as set out in the trials PLND with MIAT SOP. Cases may be referred for multidisciplinary discussion to the ATLANTA MIAT Quality Assurance Board.

### Intervention arm 2: radical therapy

In addition to SOC systemic treatment, radical therapy involves either: (1) cytoreductive radical prostatectomy, with or without PLND, or (2) prostate EBRT, with or without simultaneous PLNRT. The actual modality will be based on physician and patient preference, as well as patient comorbidities and performance status.

The surgical technique is at the discretion and expertise of the surgical team but will reflect current UK surgical practice, laid down in the cytoreductive radical prostatectomy SOP. Trial surgeons must meet minimum case volume and optimal complication outcomes prior to operating in this trial. Further, they must receive peer approval from the IP2-ATLANTA Surgeons Quality Assurance Board. For patients who are undergoing prostatectomy, no local prostate radiotherapy will be given as part of the intervention. Radiotherapy can be given subsequently for palliative reasons.

Two local prostate radiotherapy dose and fractionation options are available:

► 60 Gy in 20 fractions. Treating the prostate to 60 Gy and the seminal vesicles to 47 Gy using a simultaneous integrated boost administered over 27 days. If the pelvic lymph nodes are to be treated, then this will be done simultaneously to a dose of 47 Gy in 20 fractions (if treated).

► 74–78 Gy in 37–39 fractions. Treat the prostate to 74–78 Gy and the seminal vesicles to 60 Gy, using as simultaneous integrated boost. If the pelvic lymph nodes are to be treated, then this will be done simultaneously to a dose of 55 Gy in 37–39 fractions (in accordance with the same fractions employed for treating the prostate and seminal vesicles).

The principles of pelvic nodal treatment within the study will follow those of the PIVOTALboost study (nodal arm-B), with variation to allow both dose and fractionation regimes.[67] Quality assurance for radiotherapy will be completed by the UK national Radiotherapy Trials Quality Assurance (RTTQA) team.

### MDT: intervention arms 1 and 2

In men with low-burden disease in both intervention arms 1 and 2, MDT may be used but intent-to-use MDT is to be declared prior to randomisation. In the case of a metastatic recurrence after MDT, a re-treatment with MDT would be allowed if there were new metastatic areas/locations. The imaging reporting of metastases as well as doses and protocol for MDT will be defined and determined by the Imaging Reporting SOP and Metastases-Directed Therapy SOP.

SABR should not be delivered while concurrent chemotherapy is being delivered. Concurrent hormonal therapy is acceptable. SABR delivered to metastases must be completed within 3 months of prostate radiotherapy±PLNRT OR cytoreductive radical prostatectomy±PLND OR MIAT±PLND. Constraints on the dose and fractionations by anatomical site mirror all those defined in the SABR UK consortium guidelines V.6.1 guidelines

(2019) or, if absent, in CORE Trial Radiotherapy, Planning & Delivery guidelines V.2.0 (2018).[68 69] Quality assurance for SABR will be completed by the RTTQA body.

### Study endpoints and outcome measures

Primary and secondary endpoints for the study are presented in table 1. Study outcome measures are presented in table 2.

### Follow-Up

Follow-up will consist of 12-weekly serum PSA tests in the first year and 24-weekly thereafter, until mortality or 4 years after initial randomisation, whichever is first (table 3). PROMS will be collected every 6 months in the first year and annually thereafter, until mortality or 4 years after initial randomisation, whichever is first. Minimum follow-up for each patient will be 2 years. However, yearly follow-up will continue long-term alongside linkage to national database.

### Patient-reported outcome measures

The European Organisation for the Research and Treatment of Cancer (EORTC), Core Quality of Life Questionnaire-C30 (QLQ-C30), with prostate-specific, fatigue, elderly, general and bone metastases modules, International Prostatic Symptoms Score (IPSS), The Expanded Prostate Cancer Index Composite Bowel and Bladder (EPIC) and International Index of Erectile Function 5 (IIEF15) will be used. The EuroQol (EQ-5D-5L) will be used in the study as a generic measure of health-related quality of life which can be linked to public preferences. These data will be used to calculate quality-adjusted life-years as part of a future health economic evaluation. Patients agreeing to return questionnaires on quality of life will continue to complete quality of life data for 4 years after enrolment.

### Study visits

Follow-up visits for the administration of SOC therapy and clinical review will occur in accordance with the local hospitals' follow-up protocols. In the intervention arms, of the embedded feasibility pilot only, a prostate MRI, systematic and targeted transrectal or transperineal biopsy will be performed at 6–9 months following the initiation of the SOC therapy. In the main phase II component, these procedures can be carried out by local centres at the discretion of local clinicians and when they are, data should be collected on their outcomes.

For those randomised to MIAT or CRP/EBRT, a date for treatment(s) will be booked in accordance with the local hospital waiting lists. Removal of the urethral catheter after MIAT or CRP will occur after a minimum period of 7 days during a hospital visit or can be removed either at the General Practice (GP) surgery or at a local hospital to the patient.

Further clinical reviews will occur as SOC visits at 12, 26, 28, 32, 34 and 52 weeks in the first year and 24-weekly intervals thereafter until mortality or 2 years after enrolment, whichever is first.

| Table 1 | Study primary and secondary endpoints |
|---|---|
| Primary endpoint: Embedded feasibility pilot | 1. Recruitment, randomisation and compliance to allocation<br>2. Adverse events<br>3. Proportion of patients with pathological complete response on post-systemic therapy prostate biopsy at 6–9 months |
| Primary endpoint: Phase II | 1. Progression-free survival (PFS)<br>Defined as a composite outcome of biochemical failure; local progression; lymph node progression or bone metastases progression (new sites); or progression or development of new distant metastases, defined as lymph nodes outside the pelvis, bone or organ involvement or skeletal-related events confirmed as progression as in the STAMPEDE randomised study (Assessment of Progression; online supplemental material).[42] |
| Secondary endpoint: Phase II | 1. Adverse events and side-effect profile<br>2. Predictive factors for PFS and OS in each arm<br>3. Effect on PFS or OS from varying radiotherapy dosage and schedules<br>4. Effect on PFS and OS stratified by volume and site for local and metastatic disease<br>5. Effect on PFS and OS stratified by the use of metastases-directed therapy<br>6. Effect on PFS using an alternative definition of failure, defined as a PSA increase of >/=25% and >/=2 ng/mL if PSA was >/=2 ng/mL from baseline, or a PSA increase of >/=25% if PSA was <2 ng/mL at random assignment<br>7. Effect on PFS using an alternative definition of local progression of a soft tissue metastatic lesion: defined as an increase of >/=20% in the largest tumour dimension with a minimum absolute increase of 5 mm. Local progression of bone metastases to be assessed using MD Anderson Cancer Centre criteria with a >/=25% increase in the size of a measurable lesion on CT or a >/=25% increase in the size of ill-defined lesions on CT considered to be progression (1, 2).<br>8. Costs and resource utilisation for future cost-effectiveness analyses<br>9. In those men undergoing repeat biopsies after 6–9 months of standard of care systemic therapy, the proportion of patients with negative biopsies<br>10. In those men undergoing repeat prostate/pelvic MRI after 6–9 months of standard of care systemic therapy, the proportion of patients with a negative prostate MRI for local tumour<br>11. In those men undergoing repeat imaging (local prostate/pelvic and/or other body areas) after 6–9 months of standard of care systemic therapy, the proportion of patients with reduction on imaging of metastatic tumour deposits |

## Further follow-up imaging

Follow-up imaging to determine response from treatment on primary and metastatic disease will not be protocolled but we recommend imaging to take place when there is suspicion of progression, such as patients with a rising PSA (ie, biochemical failure). The appropriate imaging

| Table 2 | Study outcome measures |
|---|---|
| Primary outcomes: Embedded feasibility pilot | 1. Compliance to randomised arm<br>2. Recruitment and randomisation rate<br>3. Safety (adverse events)<br>4. Proportion of patients with complete pathological response on post-SOC systemic therapy prostate biopsies at 6–9 months |
| Primary outcomes: Phase II | 1. Progression-free survival (PFS) |
| Secondary outcomes: Phase II | 1. Urinary, sexual and rectal side effects<br>2. Patient-reported outcomes using validated questionnaires<br>3. Progression on PSA and imaging and impact of clinical features on progression<br>4. Health-related quality of life<br>5. Data on costs and resource utilisation for future cost-effectiveness analysis |

OS, overall survival; PSA, prostate-specific antigen.

**Table 3** Study visit schedule

| | Screening | Treatment | | Post-treatment | | | | Follow-up |
|---|---|---|---|---|---|---|---|---|
| **Visit** | **1** | **2** | **3** | **4** | **5** | **6** | **7** | **8 onwards** |
| Week (±) | 0 | 12 (±4) | 26 (±12) | 28 (±12) | 32 (±12) | 34 (±12) | 52 (±4) | Every 24 weeks (±4) (year 2–4) |
| Informed consent | X | | | | | | | |
| Inclusion and exclusion criteria | X | | | | | | | |
| Demography | X | | | | | | | |
| Medical history | X | | | | | | | |
| Vital signs/physical examination/ clinical or subject assessment | X | | X | | | | As deemed necessary based on medical history and AE review | |
| PSA blood test | X | X | X | | | X | X | X |
| PROMS questionnaires | X | | X | | | | X | X (at 24 months only) |
| Review/reporting of patient AEs/SAEs (may be performed via a face to face or telephone or email consultation) | | X | X | X | X | X | X | X |
| Blood and urine tests including those for biobanking | X | | X | | | X | At time of failure or at 24, 36, 48 months | |
| Randomisation | X* | | | | | | | |
| Standard of care therapy | X | X | X | X | X | X | X | X |
| Imaging tests (combination of but not limited to CT, X-rays, PET, MRI, bone scan) | If not already performed (SOC) | | | Recommended but not protocolled | | | As deemed necessary | |
| Prostate MRI | If not already performed (SOC) | | X Mandatory in pilot only. During main phase can be conducted at clinician discretion within standard of care process | | | | As deemed necessary | |

Continued

**Table 3** Continued

| | Screening | Treatment | Post-treatment | Follow-up |
|---|---|---|---|---|
| Biopsy | | X Mandatory in pilot only. During main phase can be conducted at clinician discretion within standard of care process | As deemed necessary | As deemed necessary |
| Testosterone blood test | X (if available) | X (recommended but not mandated) | As deemed necessary | |
| Treatment in intervention arms | | X | | |
| Removal of catheter | | X | | |

*Randomisation should be performed within 7 days from screening visit.
AEs, adverse events; PET, Positron emission tomography; PSA, prostate-specific antigen.

will be chosen as per the local hospital resources and policies. We envisage that the majority will perform a combination of a prostate MRI, bone scintigraphy, PET-CT/MRI, whole body MRI or CT chest/abdomen/pelvis.

### Long-term outcomes
Patients will be consented for their details to be linked to national registries for survival information such as NHS Information Centre/Office of National Statistics (ONS) in England/Wales and General Register Office in Scotland, Hospital Episode Statistics (HES). This ONS-HES linkage however is an optional consent.

### Statistical analyses and sample size calculation
#### Embedded feasibility and pilot
We are seeking to determine whether the randomisation of men with metastatic disease is feasible and whether men are compliant to the therapy following randomisation. We aim to approach 80 patients from up to 17 centres in the UK over a 6-month period to allow us to estimate a 33% recruitment rate with 95% CI width of approximately ±10 percentage points.

#### Main phase II component
The study will have 80% power to detect a treatment difference with an HR 0.7 in favour of any of the intervention arms compared with the control at a two-sided 5.0% significance level. This is based on the assumption that the accrual period will be uniform over 24 months, that the follow-up period will be 24 months and that the median PFS is 37 months. This calculation may be adjusted depending on the compliance rate assessed during the feasibility stage. The overall sample size will be 918 participants considering a 5% loss to follow-up (291 participants per group, 873 participants for three arms). This will allow the detection of an effect size of 9.2% increase in PFS at 24 months.

### Adverse event reporting
The Common Terminology Criteria for Adverse Events (CTCAEv5.0) domain will be used to report adverse events.[70]

### Data collection
The principal means of data collection from participant visits will be Electronic Data Capture (EDC) using the web-based InForm database. All study data will be entered into electronic Case Report Forms (eCRFs) in a database provided by the sponsor. All eCRFs will be completed using deidentified data.

### Data monitoring and archiving
A combined independent data monitoring and trial steering committee will meet twice a year. All trial documentation, including that held at participating sites and the trial coordinating centre, will be archived for a minimum of 10 years following the end of the study.

## Ethics and dissemination

This trial was approved by the Health Research Authority (HRA) Research Ethics Committee Wales (REC5; 19/WA0005). The results will be submitted for publication in peer-reviewed journals and submitted to the REC within a year of the end of the study.

## Trial funding, organisation and administration

IP2-ATLANTA trial was approved by the HRA Wales REC 5 (19/WA0005). IP2-ATLANTA is funded by the Wellcome Trust (204998/Z/16/Z). The study will be monitored periodically by trial monitors to assess the progress of the study, verify adherence to the protocol, ICH GCP E6 guidelines and other national/international requirements and to review the completeness, accuracy and consistency of the data.

## DISCUSSION

IP2-ATLANTA is a multicentre, phase II, randomised controlled trial. The study will provide level I evidence on oncological outcomes from prostate MIAT or radical therapy, in combination with MDT, against SOC treatment alone, in men with newly diagnosed hormone-sensitive metastatic prostate cancer. If either intervention arm is proven to provide significant oncological benefit, this will have wide-reaching implications on the current SOC paradigm.

## CONCLUSION

IP2-ATLANTA addresses an important research gap in the role of sequential systemic, local cytoreductive and metastasis-directed interventions in men with newly diagnosed metastatic prostate cancer.

## TRIAL STATUS

IP2-ATLANTA is open to recruitment in 13 centres in England and Wales and expected to complete its embedded feasibility pilot phase by January 2021.[71]

## Author affiliations

[1]Imperial Prostate, Division of Surgery, Department of Surgery and Cancer, Faculty of Medicine, Imperial College London, London, UK
[2]Imperial Urology, Imperial College Healthcare NHS Trust, London, UK
[3]Imperial College Clinical Trials Unit, Imperial College London, London, UK
[4]Department of Oncology, Imperial College Healthcare NHS Trust, London, UK
[5]Radiotherapy Trials Quality Assurance (RTTQA), Royal Marsden NHS Foundation Trust, London, UK
[6]Department of Oncology, Addenbrooke's Hospital, Cambridge, UK
[7]Department of Urology, Royal Devon and Exeter NHS Foundation Trust, Exeter, UK
[8]Department of Oncology, Royal Devon and Exeter NHS Foundation Trust, Exeter, UK
[9]Department of Urology, Arrowe Park Hospital, Wirral University Teaching Hospital NHS Foundation Trust, Wirral, UK
[10]Department of Clinical Oncology, Clatterbridge Cancer Centre NHS Foundation Trust, Bebington, UK
[11]Department of Oncology, Chelsea and Westminster Hospital NHS Foundation Trust, London, UK
[12]Department of Oncology, Newcastle Upon Tyne Hospitals NHS Foundation Trust, Newcastle Upon Tyne, UK
[13]Department of Oncology, London North West University Healthcare NHS Trust, Harrow, UK
[14]Department of Radiotherapy, University Hospital Southampton NHS Foundation Trust, Southampton, UK
[15]Department of Urology, Wrexham Maelor Hospital, Wrexham, UK
[16]Department of Urology, Newcastle Upon Tyne Hospitals NHS Foundation Trust, Newcastle Upon Tyne, UK
[17]Department of Urology, Northwick Park Hospital, London North West University Healthcare NHS Trust, Harrow, UK
[18]Department of Urology, Sunderland Royal Hospital, Sunderland, UK
[19]Department of Urology, Chelsea and Westminster Hospital NHS Foundation Trust, London, UK
[20]Department of Oncology, The Royal Marsden NHS Foundation and Institute of Cancer Research, London, UK
[21]Department of Urology, University Hospital Southampton NHS Foundation Trust, Southampton, UK
[22]Research, Velindre Cancer Centre, Cardiff, UK
[23]Division of Cancer and Genetics, Cardiff University School of Medicine, Cardiff, UK

**Acknowledgements** We would like to thank all the participants, study PI, trial clinicians, research nurses, Imperial Clinical Trial Unit staff and other site staff who have been responsible for setting up, recruiting participants and collecting the data for the IP2-ATLANTA trial. Further, we are grateful for the ongoing support of the Trial Management Group and our trial patient representative. Finally, we would like to thank the trial oversight provided by ICTU and our trial funder the Wellcome Trust.

**Contributors** Conception and design of the ATLANTA trial: HUA, MJC, MW, TTS, TD, AF, AS, VK, MG, NS, JS, ED, FF, NKN and ME. All authors have read and approved the final manuscript: MJC, TTS, KS, ED, JS, FF, NS, MG, AF, NKN, ME, OFN, KTJ, DP, SG, DB, GH, JM, DS, MK, AI, CB, RAP, NA, CH, IS, BR, GH, SM, BK, SM, VK, TD, JNS, MW and HUA.

**Funding** The trial was funded by the Wellcome Trust (204998/Z/16/Z).

**Competing interests** MJC's research is support by University College London Hospitals (UCLH) Charity and the Wellcome Trust. KTJ is currently supported by a research grant from the UK National Institute of Health Research (NIHR) Clinical Research Network Eastern. He has received educational and travel grants from Bayer UK, Janssen Oncology, Pfizer; Roche, Takeda. HUA's research is supported by core funding from the United Kingdom's National Institute of Health Research (NIHR) Imperial Biomedical Research Centre. HUA currently receives funding from the Wellcome Trust, Prostate Cancer UK, MRC (UK), Cancer Research UK, Sonacare Inc., Trod Medical and Sophiris Biocorp for trials in prostate cancer. HUA was a paid medical consultant for Sophiris Biocorp, Sonacare Inc. and BTG in the past 3 years.

**Patient consent for publication** Not required.

**Provenance and peer review** Not commissioned; externally peer reviewed.

**ORCID iDs**
Martin John Connor http://orcid.org/0000-0003-4033-7508
Kamalram Thippu Jayaprakash http://orcid.org/0000-0001-7217-4593
John Nicholas Staffurth http://orcid.org/0000-0002-7834-3172
Hashim Uddin Ahmed http://orcid.org/0000-0003-1674-6723

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
