## [Reviewer comments · BMJ Open]

ARTICLE DETAILS

TITLE (PROVISIONAL)	Additional Treatments to the Local tumour for metastatic prostate cancer - Assessment of Novel Treatment Algorithms (IP2-ATLANTA): Protocol for a multicentre, phase II randomised controlled trial
AUTHORS	Connor, Martin; Shah, Taimur T; Smigielska, Katarzyna; Day, Emily; Sukumar, Johanna; Fiorentino, Francesca; Sarwar, Naveed; Gonzalez, Michael; Falconer, Alison; Klimowska-Nassar, Natalia; Evans, Martin; Naismith, Olivia; Thippu Jayaprakash, Kamalram; Price, Derek; Gayadeen, Shiva; Basak, Dolan; Horan, Gail; McGrath, John; Sheehan, Denise; Kumar, Manal; Ibrahim, Azman; Brock, Cathryn; Pearson, Rachel; Anyamene, Nicola; Heath, Catherine; Shergill, Iqbal; Rai, Bhavan; Hellowell, Giles; Mccracken, Stuart; Khoubehi, Bijan; Mangar, Stephen; Khoo, Vincent; Dudderidge, Tim; Staffurth, John; Winkler, Mathias; Ahmed, Hashim

VERSION 1 – REVIEW

REVIEWER	Prabhjot Singh All India Institute of Medical Sciences, New Delhi, India
REVIEW RETURNED	02-Oct-2020

GENERAL COMMENTS	Comments; 1. As per Methodology: Standard of care: “SOC treatment as determined by the treating physician: ADT with or without Docetaxel or other systemic standard of care treatment, including but not limited to, Abiraterone Acetate or Enzalutamide. Radiotherapy to the prostate only in this arm is defined as for ‘symptom control’ in high volume (≥ 4) metastases or to mirror current accepted local radiotherapy dose regimens, external beam prostate radiotherapy using a dose of 36Gy/6Fr OR 55Gy/20Fr OR 60Gy/20Fr over 27 days as defined in Local Radiotherapy Standard Operating Procedure (SOP), for men with low volume metastases” According to author SOC means ADT would be given along with Docetaxel/ enzalutamide/Abiraterone in all patients. But according to All RCTs in castration sensitive prostate ca like CHAARTED/ STAMPEDE trials, benefits were achieved only in high volume disease patients. Still ADT with adjuvant treatment (Docetaxel/ enzalutamide/Abiraterone) are not standard for CSPC for high volume diseases. Is it the Standard of care for CSPC in UK? There is not RCT which shows Head to head comparison between Docetaxel/ enzalutamide/Abiraterone along with ADT in Castration sensitive Ca P. So what is the criteria of giving which adjuvant treatment in which patient. I think we should keep same standard of care in all patients.
---

	Why author had decided to give Local RT in standard arm? Giving prostate EBRT in SOC GROUP will confound the results. Whether it will be given to low volume or high volume is not clear. 2. As the SOC treatment would be administered prior to any local MIAT or CRP/EBRT in the intervention arms 1 and 2, an attempt at reclassifying the residual disease with a prostate MRI and biopsies would be pragmatic Do you think after hormone therapy, one can find anything on biopsy after 8-9 months? And will that change the treatment if no tumour found on follow up. 3. Trial is planning to do PLND or RT to LN in some of the patients of Ablative therapy? what are the inclusion / exclusion criteria for that? 4. How the disease will be staged in context of metastasis to measure SOC response? Bone scan or PSMA PET? 5. A proper flow chart which mentions all inclusion /exclusion criteria, procedure /tests done at each step would be great.
--	---

REVIEWER	Benjamin L Maughan Huntsman Cancer Institute, University of Utah
REVIEW RETURNED	26-Oct-2020

GENERAL COMMENTS	This is a well-written manuscript that clearly and accurately describes the rationale for this clinical trial and the present data that supports it. The only major criticism that I can foresee with the interpretation of the results from this clinical trial is related to docetaxel administration and the primary endpoint which includes PSA failure as a component of the composite endpoint. The STAMPEDE data clearly shows that OS appears to be equivalent between ADT/docetaxel and ADT/abiraterone (and by extension likely ADT/enza and ADT/apa). However, there is a clear difference in PSA-PFS and radiographic PFS between ADT/docetaxel and ADT/abiraterone. Since this trial includes any standard of care combination this very likely will influence the results unless the number of patients receiving docetaxel is the same in all arms.
---

VERSION 1 – AUTHOR RESPONSE

Reviewer: 1 Reviewer Name: Prabhjot Singh Institution and Country: All India Institute of Medical Sciences, New Delhi, India
Comments to the Author

Thank-you for taking the time to review the manuscript and for your comments which we have responded to in turn below, highlighting subsequent changes to the revised manuscript where required.

1. As per Methodology: Standard of care:

“SOC treatment as determined by the treating physician: ADT with or without Docetaxel or other systemic standard of care treatment, including but not limited to, Abiraterone Acetate or Enzalutamide. Radiotherapy to the prostate only in this arm is defined as for ‘symptom control’ in high volume (≥ 4) metastases or to mirror current accepted local radiotherapy dose regimens, external beam prostate radiotherapy using a dose of 36Gy/6Fr OR 55Gy/20Fr OR 60Gy/20Fr over 27 days as defined in Local Radiotherapy Standard Operating Procedure (SOP), for men with low volume metastases”

According to author SOC means ADT would be given along with Docetaxel/ enzalutamide/Abiraterone in all patients. But according to All RCTs in castration sensitive prostate ca like CHAARTED/ STAMPEDE trials, benefits were achieved only in high volume disease patients. Still ADT with adjuvant treatment (Docetaxel/ enzalutamide/Abiraterone) are not standard for CSPC for high volume diseases. Is it the Standard of care for CSPC in UK?

There is not RCT which shows Head to head comparison between Docetaxel/ enzalutamide/Abiraterone along with ADT in Castration sensitive Ca P. So what is the criteria of giving which adjuvant treatment in which patient. I think we should keep same standard of care in all patients.

Response 1.1: Thank you for raising these important points. For context, in the UK, NICE & NHS England considered the role of docetaxel in combination with ADT. It is now recommended docetaxel use in all men with newly diagnosed metastatic prostate cancer “who do not have significant comorbidities and have started treatment within 12 weeks of starting androgen deprivation therapy and for use six 3-weekly cycles at a dose of 75 mg/m² (with or without daily prednisolone)” (1,2). Reference to this is now included in the revised manuscript on page 13, line 392.

This decision was, in part, made following the STOpCaP meta-analysis published in the Lancet Oncology (3). During the COVID-19 pandemic, approval was provided to use enzalutamide and/or abiraterone acetate as an alternative. Therefore, the IP2-ATLANTA trial systemic therapy design is based on current accepted UK practice. In this trial we estimate 90% of men will have one agent in addition to ADT.

The decision to allow all available systemic agents in this trial is also supported by a recent meta-analysis comparing anti-androgen compounds against docetaxel by Marchioni and colleagues (4). This meta-analysis reported statistically insignificant lower overall mortality rates with abiraterone (HR 0.89, 95% CI 0.76 p=1.05), enzalutamide (HR 0.90, 95% CI 0.69 p = 1.19) and apalutamide (HR 0.90, 95% CI 0.67 p = 1.22) when compared to docetaxel (4). As such, we do not have concerns that alternative anti-androgen compounds would lead to inferior, or indeed relevant superior, outcomes compared to docetaxel administration in addition to ADT.

This pragmatic trial design permitting a range of upfront systemic agents reflects other trial designs within this space (e.g. NCT03655886, NCT03298087, NCT03456843). This design allows the trial to continue despite real world changes in SOC practice, which in this disease space are happening swiftly.

In the IP2-ATLANTA study men will be stratified at randomisation by “ADT alone” or “ADT + additional systemic agent” to help mitigate against trial arm bias. Furthermore, we have provided a strict treatment window to ensure exposure of ADT and any additional systemic agent is comparable in each arm.

- (1) NICE. Prostate cancer: diagnosis and management NICE guideline [NG131]. 2019. Available at: <https://www.nice.org.uk/guidance/ng131>
- (2) NHS England. Clinical Commissioning Policy Statement: Docetaxel in combination with androgen deprivation therapy for the treatment of hormone naïve metastatic prostate cancer. 2016. Available at: <https://www.england.nhs.uk/wp-content/uploads/2016/01/b15psa-docetaxel-policy-statement.pdf>
- (3) Vale CL, Burdett S, Rydzewska LHM, et. al., on behalf of the STOpCaP Steering Group (2015). Addition of docetaxel or bisphosphonates to standard of care in men with localised or metastatic, hormone-sensitive prostate cancer: a systematic review and meta-analyses of aggregate data. *Lancet Oncology* S1470-2045 (15) 00489-1.
- (4) Marchioni, M, Di Nicola, M, Primiceri, G, et al., 2020. New antiandrogen compounds compared to docetaxel for metastatic hormone sensitive prostate cancer: results from a network meta-analysis. *The Journal of Urology*, 203(4), pp.751-759.

Why author had decided to give Local RT in standard arm? Giving prostate EBRT in SOC GROUP will confound the results. Whether it will be given to low volume or high volume is not clear.

Response 1.2: Following the results from the STAMPEDE trial's pre-specified subgroup (CHAARTED definition low burden disease) where a significant OS was reported (3-year OS 81% vs 73%, HR 0.68, [95% CI 0.52–0.90]; p=0.007), our healthcare regulatory authority, NHS England, has allowed for the delivery of local prostate radiotherapy in low burden men only within standard care (5).

As such, IP2-ATLANTA will mirror this guideline and offer low burden men only the current standard of care radiotherapy option within the trial. Our pragmatic trial design will be more acceptable by patients and their physicians and reflects our ethical responsibility in permitting access to life prolonging standard treatment in the control arm.

Patients with high volume disease will not receive local prostate radiotherapy in any form in the control arm. Further, no patients in this arm will not be able to access the radical dose/fractionations directed to the prostate and pelvic lymph nodes that are on offer in the radiotherapy/surgery arm in our RCT, if they decided to participate and consent to randomisation. Furthermore, they will not be able to access Metastasis-Directed Therapy. Our study compares the use of systemic therapy and some men getting local prostate radiotherapy at a low dose (standard care) to dose escalation forms of radiotherapy or surgery to the prostate and metastases where appropriate. This is contained in the revised manuscript on page 12, line 399 and in Figure 1.

(5) Clinical Commissioning Policy: External beam radiotherapy for patients presenting with hormone sensitive, low volume metastatic prostate cancer at the time of diagnosis [P200802P] (URN: 1901). 2020. Available at: <https://www.england.nhs.uk/wp-content/uploads/2020/11/1901-Policy.pdf>

2. "As the SOC treatment would be administered prior to any local MIAT or CRP/EBRT in the intervention arms 1 and 2, an attempt at reclassifying the residual disease with a prostate MRI and biopsies would be pragmatic".

Do you think after hormone therapy, one can find anything on biopsy after 8-9 months? And will that change the treatment if no tumour found on follow up.

Response 1.3: Thank you for raising this point. It is unknown whether residual tumour foci will be detected in patients post-systemic therapy. Drawing on studies from patients with high-risk non-metastatic prostate cancer who underwent ADT with or without docetaxel/abiraterone acetate prior to radical prostatectomy, the rate of complete local pathological response is estimated between 0% and 10% (6-8). Higher rates of complete local pathological response in the embedded feasibility phase

would lead to the continuation of mandatory biopsies, to enable men to avoid unnecessary local treatment and its associated toxicity. This would, however, only be introduced if patients found the second biopsy acceptable.

Furthermore, there is some evidence that post-systemic therapy biopsy features may have a role in prognostication of aggressive disease. This will also be evaluated in the pilot phase and in the main phase if the biopsy is continued (8,9). These supporting citations now accompany the text on page 14, line 425.

(6) Köllermann, J., Caprano, J., Budde, A., et al., 2003. Follow-up of nondetectable prostate carcinoma (pT0) after prolonged PSA-monitored neoadjuvant hormonal therapy followed by radical prostatectomy. *Urology*, 62(3), pp.476-480.

(7) Bream, M.J., Dahmouh, L. and Brown, J.A., 2013. pT0 Prostate Cancer: predictive clinicopathologic features in an American population. *Current urology*, 7(1), pp.14-14.

(8) McKay RR, Ye H, Xie W et al. Evaluation of intense androgen deprivation before prostatectomy: a randomized phase II trial of enzalutamide and leuprolide with or without abiraterone. *Journal of Clinical Oncology*. 2019 Apr 10;37(11):923.

(8) O'Brien, C., True, L.D., Higano, C.S., Rademacher, B.L., Garzotto, M. and Beer, T.M., 2010. Histologic changes associated with neoadjuvant chemotherapy are predictive of nodal metastases in patients with high-risk prostate cancer. *American journal of clinical pathology*, 133(4), pp.654-661.

(9) Evans, A.J., 2018. Treatment effects in prostate cancer. *Modern Pathology*, 31(1), pp.110-121.

3. Trial is planning to do PLND or RT to LN in some of the patients of Ablative therapy? what are the inclusion / exclusion criteria for that?

Response 1.4: PLND will be performed based on presence of resectable disease, patient fitness and consent/discussion with operating surgeon, as set out in the trials PLND with MIAT SOP. This has now been included in the revised manuscript on page 14, line 436.

4. How the disease will be staged in context of metastasis to measure SOC response? Bone scan or PSMA PET?

Response 1.5: Conventional imaging (CT TAP / Bone Scan) recommended following systemic therapy and prior to local treatment. In the embedded feasibility phase all men will undergo a repeat prostate MRI in the experimental arms prior to local treatment. This is set out in Table 3 in the revised manuscript.

5. A proper flow chart which mentions all inclusion /exclusion criteria, procedure /tests done at each step would be great.

Response 1.6: The inclusion/exclusion criteria and an overview of the procedures in each arm is set out in Figure 1. Table 3. These we believe provide an in-depth study schedule with all tests and procedures at each trial visit.

Reviewer: 2

Reviewer Name: Benjamin L Maughan

Institution and Country: Huntsman Cancer Institute, University of Utah

Comments to the Author

This is a well-written manuscript that clearly and accurately describes the rationale for this clinical trial and the present data that supports it.

Thank you for taking the time to review the manuscript and for your positive comments.

The only major criticism that I can foresee with the interpretation of the results from this clinical trial is related to docetaxel administration and the primary endpoint which includes PSA failure as a component of the composite endpoint. The STAMPEDE data clearly shows that OS appears to be equivalent between ADT/docetaxel and ADT/abiraterone (and by extension likely ADT/enza and ADT/apa). However, there is a clear difference in PSA-PFS and radiographic PFS between ADT/docetaxel and ADT/abiraterone. Since this trial includes any standard of care combination this very likely will influence the results unless the number of patients receiving docetaxel is the same in all arms.

Response 2.1: Thank you for raising this important point. In the IP2-ATLANTA study, patients will be stratified at randomisation by “ADT alone” or “ADT + additional systemic agent” to help mitigate against trial arm bias; this will ensure equal numbers of patients having additional agents to ADT in each arm. Furthermore, we have provided a strict treatment window to ensure exposure of ADT and any additional systemic agent are comparable in each arm.

For context, in the UK, NICE & NHS England considered the role of docetaxel in combination with ADT. It is now recommended docetaxel use in all men with newly diagnosed metastatic prostate cancer “who do not have significant comorbidities and have start treatment within 12 weeks of starting androgen deprivation therapy and for use six 3-weekly cycles at a dose of 75 mg/m² (with or without daily prednisolone)” (1,2). Reference to this is now included in the revised manuscript on page 13, line 392.

This decision was, in part, made following the STOpCaP meta-analysis published in the Lancet Oncology (3). During the COVID-19 pandemic approval has been provided to use Enzalutamide and/or Abiraterone Acetate as alternatives. Therefore, the IP2-ATLANTA trial systemic therapy design is based on current accepted UK practice. In this trial we estimate 90% of patients will have an agent in addition to ADT.

The decision to allow all available systemic agents in this trial is also supported by a recent meta-analysis comparing anti-androgen compounds against docetaxel by Marchioni and colleagues (4). This meta-analysis reported a non-statistically significant lower overall mortality rates with abiraterone (HR 0.89, 95% CI 0.76 p=1.05), enzalutamide (HR 0.90, 95% CI 0.69 p = 1.19) and apalutamide (HR 0.90, 95% CI 0.67 p = 1.22) when compared to docetaxel (4). As such, we do not have concerns that alternative anti-androgen compounds would lead to inferior, or clinically relevant superior, outcomes compared to Docetaxel administration.

However, we accept that there may be subtle benefits of each agent in contributing to the composite progression-free survival endpoint (PSA or radiological). We would suggest that these would be outweighed by the “additive” impact of the allocated local and metastasis-directed trial treatments on overall progression-free survival composite, should these indeed prove to be significant in each study arm.

This pragmatic trial design permitting a range of upfront systemic agents reflects other trial designs within this space (e.g. NCT03655886, NCT03298087, NCT03456843). This design also allows the trial to continue despite real world changes in SOC practice. Fixing the SOC arm would lead to lack of equipoise by oncology colleagues in the UK and recruitment would falter significantly. Our design does also allow for future-proofing against what can often be unfair criticisms against RCTs which report at a time when SOC has changed, with many of our colleagues then arguing the RCT is no longer relevant.

- (1) NICE. Prostate cancer: diagnosis and management NICE guideline [NG131]. 2019. Available at: <https://www.nice.org.uk/guidance/ng131>
- (2) NHS England. Clinical Commissioning Policy Statement: Docetaxel in combination with androgen deprivation therapy for the treatment of hormone naïve metastatic prostate cancer. 2016. Available at: <https://www.england.nhs.uk/wp-content/uploads/2016/01/b15psa-docetaxel-policy-statement.pdf>
- (3) Vale CL, Burdett S, Rydzewska LHM, et. al., on behalf of the STOpCaP Steering Group (2015). Addition of docetaxel or bisphosphonates to standard of care in men with localised or metastatic, hormone-sensitive prostate cancer: a systematic review and meta-analyses of aggregate data. *Lancet Oncology* S1470-2045 (15) 00489-1.
- (4) Marchioni, M, Di Nicola, M, Primiceri, G, et al., 2020. New antiandrogen compounds compared to docetaxel for metastatic hormone sensitive prostate cancer: results from a network meta-analysis. *The Journal of Urology*, 203(4), pp.751-759.

Reviewer: 1

Competing interests 1: None Declared

Reviewer: 2

Competing interests 1: None declared

VERSION 2 – REVIEW

REVIEWER	Prabhjot Singh All India Institute of Medical Sciences, New Delhi, India
REVIEW RETURNED	21-Jan-2021

GENERAL COMMENTS	Thanks for replying to all queries satisfactorily
---

REVIEWER	Benjamin L. Maughan Huntsman Cancer Institute, University of Utah, USA
REVIEW RETURNED	26-Jan-2021

GENERAL COMMENTS	This trial is addressing a very important need in prostate cancer today. The authors appropriately discuss the recent publication from the STAMPEDE trial demonstrating an improved OS in patients with low-volume disease. This highlights the potential benefit though unproven in the era of intensified therapy. This trial further expands on that rationale by exploring multiple methods of treatment including surgery, radiation or focal cyrotherapy. This is a very pertinent trial and given its ambitious size will be sufficient to address these important clinical questions. I strongly recommend approval and publication of this manuscript.
---